# From Data Deluge to Data Curation: A Filtering-WoRA Paradigm for Efficient Text-based Person Search

Submission Id: 24

## ABSTRACT

In text-based person search endeavors, data generation has emerged as a prevailing practice, addressing concerns over privacy preservation and the arduous task of manual annotation. Although the number of synthesized data can be infinite in theory, the scientific conundrum persists that how much generated data optimally fuels subsequent model training. We observe that only a subset of the data in these constructed datasets plays a decisive role. Therefore, we introduce a new Filtering-WoRA paradigm, which contains a filtering algorithm to identify this crucial data subset and WoRA (Weighted Low-Rank Adaptation) learning strategy for light fine-tuning. The filtering algorithm is based on the cross-modality relevance to remove the lots of coarse matching synthesis pairs. As the number of data decreases, we do not need to fine-tune the entire model. Therefore, we propose a WoRA learning strategy to efficiently update a minimal portion of model parameters. WoRA streamlines the learning process, enabling heightened efficiency in extracting knowledge from fewer, yet potent, data instances. Extensive experimentation validates the efficacy of pretraining, where our model achieves advanced and efficient retrieval performance on challenging real-world benchmarks. Notably, on the CUHK-PEDES dataset, we have achieved a competitive mAP of 67.02% while reducing model training time by 19.82%.

## CCS CONCEPTS

• **Information systems** → **Retrieval efficiency**; **Retrieval effectiveness**; **Multimedia and multimodal retrieval**.

## KEYWORDS

Text-based Person Search, Data-centric Learning, Low-Rank Adaptation, Visual-language Pre-training

## 1 INTRODUCTION

Compared to traditional image-based person search [57, 64, 65, 82, 86], which seeks to retrieve target individuals from a vast array of footage or images across different locations and times, text-based person search locates interested individuals from a pool of candidates based on pedestrian descriptions [36]. Given that pedestrian image queries may not be available, text-guided person search emerges as an alternative method. The key lies in mining the fine-grained information from images and texts, blending the complexity of natural language processing with the subtle nuances of visual recognition, and establishing their correspondence. By leveraging the ability to understand complex human descriptions and accurately identify and retrieve images of individuals from a camera system, it can be applied to broad applications in public safety domains such as missing person searches [2, 59], and rescue operations [56]. As the sub-task of vision-language retrieval, text-based person search models usually require extensive data for training,

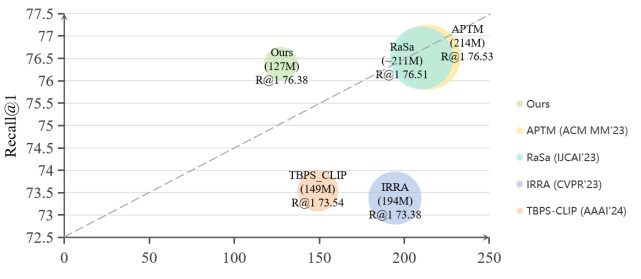

**Figure 1: Comparison between the proposed method and existing approaches in terms of Recall@1 and the parameter numbers. We observe that our method deploys fewer parameters while achieving a competitive Recall@1, *i.e.*, APTM [73], RaSa [4], IRRA [25] and TBPS-CLIP [5].**

where, however, the number of pedestrian data is limited. Most datasets [13, 37, 73, 86] are constructed from three sources. (1) The first source is through sampling from camera footage, accompanied by manual annotations. However, constructing large-scale datasets is often infeasible due to privacy concerns and high costs. (2) The second source involves collecting images and short videos from the internet. Despite expanding dataset sizes, the noisy web text and the inconsistent quality of task images are generally suboptimal for fine-grained vision-language learning. (3) Therefore, most researchers [73] resort to leverage the generative models, *e.g.*, GAN [84] and Diffusion [50, 55]. For instance, APTM [73] has introduced 1.51M image-text pairs generated by Stable Diffusion [50], showcasing the potential of training on a large synthesized dataset.

Despite significant progress in learning from large synthesized datasets, a fundamental challenge persists: **how can we efficiently extract knowledge when faced with an effectively infinite amount of data, considering the substantial computational costs incurred during training?** We observe two key points: (1) Previous works [73, 81, 84] indicate that performance gains diminish even with abundant generated data. This suggests that not all information in massive synthetic datasets is equally useful; instead, a well-chosen subset, or coreset, may be sufficient to capture the essential training information. (2) When learning from a coreset, updating the entire parameter space is unnecessary to achieve training precision. This insight opens the door to significantly reducing both training time and model complexity. Thus, our focus shifts to developing strategies that achieve high accuracy with reduced data and a compact parameter footprint.

To this end, we propose a new Filtering-WoRA paradigm, which contains a novel two-stage data filtration method aimed at identifying the coreset to enhance model performance and a WoRA (Weighted Low-Rank Adaptation) algorithm to optimize the pretraining and fine-tuning models, enabling training with fewer parameters while maintaining model performance and increasing

computational speed. Specifically, our process begins with dataset purification, including the synthesized dataset for pre-training and the real-world dataset for fine-tuning. To filter out low-quality image-text pairs, *e.g.*, incomplete descriptive details or blurred image details, we leverage the off-the-shelf large cross-modality model to extract features from both images and texts within the dataset and then calculate the cosine similarity between projected image embeddings and projected text embeddings. This process yields a similarity score for each image-text pair, facilitating the selection of high-quality datasets based on our predetermined threshold. Subsequently, to reduce model parameters and enhance computational speed, we opt to freeze the weights from pre-training, indirectly training some dense layers in the neural network by optimizing rank decomposition matrices that change during the adaptation process. By decomposing pre-training and fine-tuning weights into magnitude and direction, our WoRA method introduces three new dimensions to facilitate the modification of the weight matrix and rank decomposition matrix. This approach allows for learning a minimal amount of parameters while simultaneously boosting model performance (see Figure 1). In summary, our contributions are:

- We introduce a new Filtering-WoRA paradigm for efficient text-based person search, which streamlines learning and improves efficiency through focused data curation and targeted parameter updates. The filtering algorithm targets relevant, high-quality synthesized data by assessing cross-modality relevance, while WoRA (Weighted Low-Rank Adaptation) enables lightweight fine-tuning of a minimal set of model parameters.
- Extensive experiments on three widely-used benchmarks verify that our method could save 19.82% training time compared with the vanilla baseline, while also achieving competitive 76.37%, 66.65%, 67.90% Recall@1 accuracy on CUHK-PEDES, RSTPReid and ICFG-PEDES, respectively.

## 2 RELATED WORK

**Vision-Language Pre-training.** Current VLP research predominantly bifurcates into coarse-grained and fine-grained methodologies. Coarse-grained approaches employ convolutional networks [23, 24, 26, 77] or visual Transformers [27, 28, 34, 45, 49] to extract and encode holistic image features, thereby constructing vision-language models (VLMs). Techniques such as SOHO [23] propose leveraging a Visual Dictionary (VD) for the extraction of comprehensive yet compact image features, facilitating enhanced cross-modal comprehension. ALBEF [34] introduces a contrastive loss for aligning image and text representations before their fusion through cross-modal attention, fostering more grounded vision-language representation learning. Additionally, it utilizes momentum distillation to augment learning capabilities from noisy network data. Although these holistic image-focused methods are efficient, their performance is generally outpaced by fine-grained approaches. Inspired by advancements in the NLP domain, fine-grained methods [10, 15, 31, 35, 40, 47, 60] employ pre-trained object detectors [3, 54] trained on annotated datasets of common objects, such as COCO [41] and Visual Genome [29]. This enables models to recognize and classify all potential object regions within images, representing them as a collection of object-centered features. For

instance, VinVL [76] enhances visual representations for V+L tasks and develops an improved object detection model to provide object-centered image representations. However, this object-centered feature representation struggles to capture relationships between multiple objects across different regions, limiting its effectiveness in encoding multi-granularity visual concepts. Another limitation is the inability of the object detector to recognize uncommon objects not present in the training data. Recently, novel approaches have emerged to bridge the learning of object-level and image-level alignments. E2E-VLP [71] employs DETR [6] as the object detection module to enhance detection capabilities. KD-VLP [44] relies on external object detectors for object knowledge distillation, facilitating cross-modal alignment learning across different semantic layers. OFA [62] formulates visual-linguistic tasks as a sequence-to-sequence (seq2seq) problem, adhering to instruction-based learning during both pre-training and fine-tuning phases, eliminating the need for extra task-specific layers. Uni-Perceiver [87] constructs a unified perception architecture, using a single Transformer and shared parameters for diverse modes and tasks, employing a non-mixed sampling strategy for stable multi-task learning. X-VLM [74] and X2-VLM [75] propose an integrated model with a flexible modular architecture to simultaneously learn multi-granularity alignment and localization, achieving the capability to learn infinite visual concepts related to various text descriptions. In this work, we leverage the proficient vision-language pre-trained model to filter noisy data.

**Text-Image Person Search.** Based on the challenging task of language-based person search, which is a fine-grained, cross-modal retrieval challenge, a significant number of methodologies have been developed in recent years to tackle this issue. Existing approaches can broadly be classified into two categories: those based on cross-modal attention interaction [37, 57, 58, 65] and those without cross-modal attention interaction [9, 13, 64, 73, 83]. Methods leveraging cross-modal attention interactions facilitate correspondences between image regions and textual phrases by pairing inputs and predicting image-text matching scores through attention mechanisms. This enhances interaction between modalities, effectively bridging the modality gap, albeit at the cost of increased computational complexity. For instance, Li *et al.* [37] propose a recurrent neural network with gated neural attention to enhance cross-modal learning. Shao *et al.* [57] introduce a multimodal shared dictionary (MSD) to reconstruct visual and textual features, employing shared, learnable archetypes as queries. To improve person search performance, features for both modalities are extracted in the feature space with uniform granularity, ensuring semantic consistency. Conversely, methods without cross-modal attention interaction, through the construction of diverse model structures and objective functions, align representations of the two modalities within a shared feature space. These lightweight models, not reliant on complex cross-modal interactions, are computationally more efficient and have even achieved better results than their attention-based counterparts. Zheng *et al.* [83] build an end-to-end dual-path convolutional network to learn image and text representations to take full advantage of supervision capabilities. However, all the person search methods mentioned above fine-tune the entire network for high accuracy, which inherently slows down the process. In

contrast, we propose a method based on cross-modal feature extraction for rapid candidate selection and data ranking scores. This approach aims to mitigate the impact of low-quality text-image pairs while reducing model computational parameters to maintain high efficiency in processing.

**Data-Centric Learning.** With the surge in popularity of large language models, an increasing demand for vast datasets for model training has emerged [52, 73–75]. However, open-source datasets constructed for training models in real-world scenarios, such as the MALS dataset [73], may encounter issues like incorrect text descriptions, poor image or text quality, and insufficient feature matching between image-text pairs, all of which can adversely affect model training performance. As the size of datasets expands, it is observed that their quality does not invariably scale in tandem [84]. Frequently, a subset of high-caliber data can attain or even exceed the utility of a voluminous but qualitatively inferior dataset. This phenomenon underscores the paramount importance of meticulously curating high-quality datasets, thereby highlighting the necessity for efficiency and precision in dataset construction. For instance, data selection methods [22, 51, 69] aim to identify and train only with the most relevant and informative examples, discarding irrelevant or redundant data. This leads to more efficient learning and improved model performance, especially when dealing with large datasets. Active learning seeks to reduce labeling costs by selecting the most informative instances for annotation. Data cleaning and preprocessing techniques aim to remove noise, errors, and inconsistencies from the data, making it more suitable for learning. Coreset selection [32, 33] focuses on identifying a small subset of data points (a core set) sufficient for training a model that performs well across the entire dataset. By selecting a representative subset, coreset selection can reduce the computational costs of training and improve model generalization. Our approach advocates for a coreset method aimed at enhancing model performance, proposing the deploy of the off-the-shelf visual-language models to segment and filter the dataset for text and image pair matching scores, thereby obtaining a coreset dataset for effective training.

## 3 METHODOLOGY

### 3.1 Baseline Revisit

We do not pursue the network contribution in this work, but focus on the training efficiency. Our method can be adopted to most existing works. Without loss of generality, we apply the widely-used baseline, APTM [73], to simplify the illustration as well as a fair comparison with other methods. In particular, the framework comprises three encoders, *i.e.*, image encoder, text encoder, and cross encoder, along with two MLPs-based headers. The entire training process contains two phases, *i.e.*, pre-training on the synthesized dataset and fine-tuning on the downstream datasets. As shown in Figure 2, the [CLS] embedding represents the aggregated feature of the image / text from the image encoder and the text encoder respectively. The cross encoder integrates image and text representations for prediction tasks, leveraging the latter 6 layers of Bert to process the previously obtained text and image embeddings, thereby discerning their semantic relationship. We adopt two types of loss functions to bolster alignment constraints, tailored for both image-text and image-attribute associations. The image-text functions encompass

Image-Text Contrastive Learning (ITC), Image-Text Matching Learning (ITM), and Masked Language Modeling (MLM), while Attribute Prompt Learning contains Image-Attribute Contrastive Learning (IAC) loss, Image-Attribute Matching Learning (IAM) Loss, and Masked Attribute Modeling (MAM) Loss. The overall APL loss is $\mathcal{L}_{APL} = \frac{1}{3} \left( \mathcal{L}_{iac} + \mathcal{L}_{iam} + \mathcal{L}_{mam} \right)$, and the full pre-training loss is formulated as: $\mathcal{L}_{total} = \mathcal{L}_{itc} + \mathcal{L}_{itm} + \mathcal{L}_{mlm} + \eta \mathcal{L}_{APL}$, where $\eta$ is empirically set as 0.8 following [73].

### 3.2 Data Filtering

As highlighted in our introduction, the acquisition of images is challenged by high annotation costs and concerns over individual privacy and security, necessitating the generation of a large volume of image-text pairs $(I, T)$ through diffusion models, complemented by text descriptions generated by large language models [73]. Despite the potential for achieving high accuracy through extensive pretraining on such data, it becomes apparent that not all generated data are equally effective, with a significant portion being redundant. Additionally, these synthesized datasets often include noise in the form of image-text pairs with poor matching quality, which can inadequately represent the visual content of images. Such pairs serve as suboptimal signals for learning fine-grained visual-language alignment, potentially disrupting model training. This observation led us to ponder whether reducing the volume of data, focusing solely on a core set, could suffice for model training. Consequently, we devised a data filtering solution a data filtering approach based on the off-the-shelf large cross-modality model, *e.g.*, BLIP-2 [32], leveraging its efficient multimodal feature extraction capabilities to isolate features from both text and images (see Figure 3). We intend to find the appropriate filtering methodology and thresholds to isolate a high-quality core set.

Our method is executed in two phases, starting with the filtering of the synthesized dataset, *i.e.*, MALS, used for pretraining the model. Initially, all image-text pairs $(I_M, T_M)$ within the dataset are processed. Given an image-text pair, the large cross-modality model, *i.e.*, BLIP-2, is employed to extract the corresponding image feature $f_{I_M}$ and text feature $f_{T_M}$. Subsequently, the training set captions from the real-world downstream dataset, *i.e.*, CUHK-PEDES, acting as distractor texts $T_C$, are utilized. For each pair, we calculate the cosine similarity between projected image embeddings and projected text embeddings to ascertain their self-similarity $Sim_{self}$. Simultaneously, the distractor similarity $Sim_{other}$ is determined by calculating the similarity between the image feature $f_{I_M}$ and a randomly selected subset of 10,000 distractor texts $T_C$. The following is the formula for calculating the similarity:

$$Sim_{self} = cos\left(f_{I_M}, f_{T_M}\right), Sim_{other} = cos\left(f_{I_M}, f_{T_C}\right), \quad (1)$$

where $cos(,)$ denotes the cosine similarity. After computing these two types of similarities for all image-text pairs within the training dataset, and then we sort texts according to the similarity from highest to lowest, we set a ranking threshold as 50. It means that an image-text pair is retained only if its self-similarity ranks within the top 50 of all calculated similarities. Through this threshold-based filtering, we discarded 21% of low-quality image-text pairs, retaining 79% to form a new pretraining dataset, Filtered-MALS.

Similarly, we also could filter the CUHK-PEDES training set in the fine-tuning phase to remove the noisy pair in the real-world

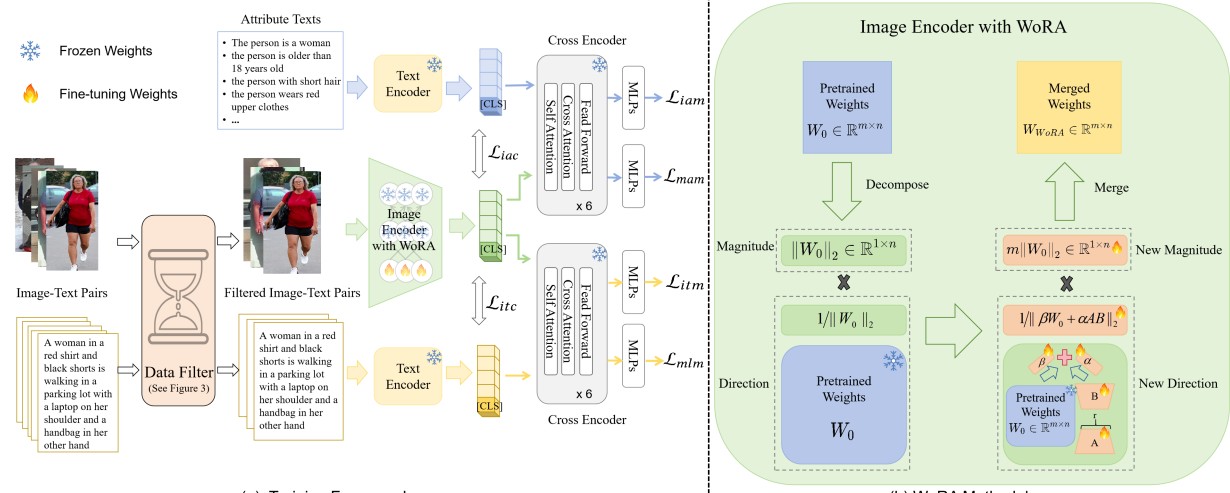

(a) Training Framework

(b) WoRA Methodology

**Figure 2: An overview of our framework. (a) shows the flow chart of the entire training pipeline. We first obtain the filtered training image-text pairs. Then we augment the text as attribute texts according to keywords. We extract the corresponding features through image encoder, text encoder and cross encoder. There are six loss objectives of both text-image and attribute-image matching tasks. (b) is an in-depth illustration of WoRA methodology, meticulously applied within the context of an image encoder. The model is updated by fine-tuning the decomposition of the pre-trained weights into amplitude and direction components and updating both components using LoRA [21] while adding the $\alpha$ and $\beta$ learnable parameters. Since the image encoder consumes most GPU memory and time. In practice, we mainly apply the WoRA on the image encoder.**

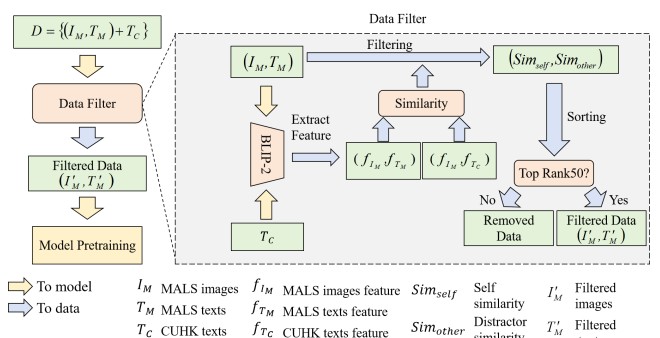

**Figure 3: An overview of our data filtering process. We first employ Blip-2 [32] to extract features from the input image-text pair $(I, T)$ and the distractor text $T_C$. Next, we compute the similarity and rank the results accordingly, ultimately generating the filtered dataset.**

training set. Employing the same calculation as in the first phase, we extract image features $f_{I_C}$ and text features $f_{T_C}$ for all pairs $(I_C, T_C)$. Each image calculates the cosine similarity with both the ground-truth text and a subset of 10,000 random distractor texts $f_{T_{C'}}$. Similarly, the similarity can be formulated as:

$$Sim_{self} = cos\left(f_{I_C}, f_{T_C}\right), Sim_{other} = cos\left(f_{I_C}, f_{T_{C'}}\right). \quad (2)$$

Upon computing and ranking the similarities across the entire training set, we set a relatively loose ranking threshold of 1800, given that most pairs are human-annotated. This selection process results in the removal of 10% low-quality image-text pairs, leaving 90% to form a refined dataset, Refined-CUHK, for model fine-tuning.

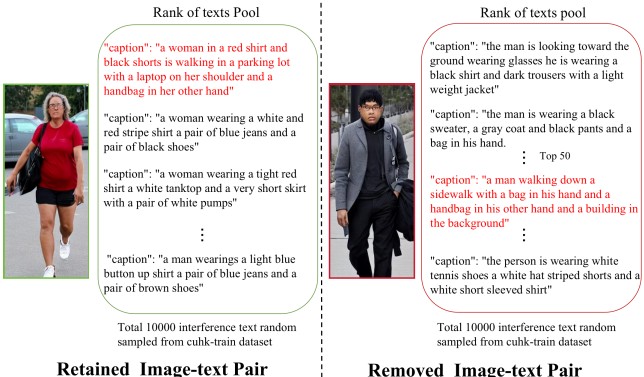

**Figure 4: Visual explanation of data filtering. The part on the left of the image shows the high-quality image retained after our screening strategy and its corresponding red text description, while the person image on the right represents the low-quality image text pairs that are filtered out beyond the threshold, *i.e.*, top50. We deploy the real-world training set as distractors to filter low-relevance synthesized image-text pairs according to the similarity since there are no overlaps.**

**Discussion. The mechanism of filtering.** The motivation is within the spectrum of generated datasets. (1) It is worth noting that not all contents bear relevance, with a portion comprising low-quality image-text pairs. Such instances of subpar alignment between textual descriptions and corresponding images can invariably exert a deleterious impact, compromising the model training.

Predominantly, the segments of the dataset that contribute most significantly to the performance of the model are those encompassing high-quality data, often referred to as the 'coreset'. (2) The extensive volume of data implicated in the pretraining phase incurs substantial computational costs, necessitating considerable resources in terms of computational power and temporal investment. Thus, data filtering provides two primary advantages: (1) Our filtering algorithm identifies the core set of image-text pairs that are most crucial for improving data quality during model training. This selective process ensures the model is exposed to higher-quality data, enhancing its learning capabilities. (2) Retaining only the core set significantly reduces the dataset size, thereby decreasing computational overhead during training. This results in reduced computational requirements and training time. Figure 4 illustrates an example of the data filtering process.

### 3.3 Weighted Low-Rank Adaptation

While the introduction of a pretrain-finetune paradigm to train models for person search tasks has achieved commendable results [73], the expansion in model and dataset sizes significantly increases the number of parameters to be trained, demanding substantial computational resources and sacrificing training efficiency. To address this issue, several parameter-efficient fine-tuning methods [20] have been proposed, aiming to fine-tune pretrained models using the minimum number of parameters. Among these, LoRA [21] has gained popularity due to its simplicity and efficacy. LoRA employs a low-rank decomposition for the pretrained weight matrix $W_0 \in \mathbb{R}^{m \times n}$, $W_0 + \triangle W = W_0 + BA$, where $B \in \mathbb{R}^{m \times r}$, $A \in \mathbb{R}^{r \times n}$, rank $r \ll \min(m, n)$. $\triangle W$ is adjusted by $\frac{8}{r}$. Inspired by the LoRA [21], which updates only a small part of the model weight to improve efficiency, DoRA [43] decomposes the weight into two parts: direction and amplitude. DoRA improves the adaptability and efficiency of the model, which can be formulated as $W_{DoRA} = m \frac{W_0 + BA}{\|W_0 + BA\|_2}$, where $m \in \mathbb{R}^{1 \times n}$ is the magnitude vector. $\|\cdot\|_2$ denotes the L2 norm of a matrix across each column.

However, these two methods still limit the parameter freedom. Therefore, we introduce the Weighted Low-Rank Adaptation (WoRA) model to address the capacity gap still present between LoRA [21] and fine-tuning (see Figure 2 right). Drawing from the DoRA [43] approach, which reparameterizes model weights into magnitude and direction components for fine-tuning, WoRA introduces new learnable parameters, $\alpha$ and $\beta$, to facilitate parameter learning. Given that pretrained weights already possess a vast repository of knowledge suitable for various downstream tasks, we configure these learnable parameters to ensure the model acquires sufficient capability in both magnitude and direction. This allows for the model to adapt to downstream tasks by updating parameters that exhibit significant magnitude or directional changes. The formal representation of our WoRA model is:

$$W_{WoRA} = m \frac{\beta * W_0 + \alpha * BA}{\|\beta * W_0 + \alpha * BA\|_2}. \tag{3}$$

In our model, parameters denoted with an overline represent trainable parameters. The proposed Weighted Low-Rank Adaptation (WoRA) demonstrates learning capabilities comparable to full fine-tuning. During inference, WoRA integrates with pretrained weights,

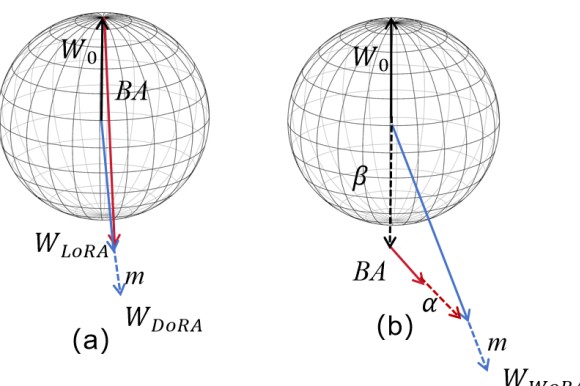

**Figure 5: Intuitive comparison of LoRA, DoRA, and our proposed WoRA. (a) Here we show a common case during optimization, *i.e.*, negative correlation against $W_0$, which both LoRA and DoRA are struggling with. The bias parameter $BA$ is hard to learn, considering the weight decay and other regularization. (b) In contrast, we deploy two float scalars in WoRA, *i.e.*, $\alpha$ and $\beta$, which could efficiently adjust the vector and provide better flexibility.**

introducing no additional latency while enhancing both learnability and computational efficiency. Moreover, by incorporating new learnable parameters, WoRA achieves superior performance compared to LoRA [21] and DoRA [43].

**Discussion. Why is WoRA better than LoRA and DoRA?** Both DoRA and LoRA are specific cases of the proposed WoRA. Specifically, DoRA can be derived from WoRA by setting $\alpha = 1$ and $\beta = 1$ as fixed constants. Similarly, setting $\alpha = 1$, $\beta = 1$, and $m = 1$ yields the identical behavior as LoRA. Consider a scenario involving negative correlation, which is common during training. As illustrated in Figure 5 (a), both LoRA and DoRA struggle with the typical negative correlation against the original weight $W_0$, necessitating **the learning of a large $BA$—a challenging and unstable process due to weight decay and other regularization terms**. In contrast, by introducing two scalar parameters, $\alpha$ and $\beta$, WoRA provides a more flexible and adaptive space for optimization. Our method can readily achieve negative correlation by setting a negative $\beta$, as depicted in Figure 5 (b). Additionally, $\alpha$ offers fine-grained control over $BA$ during optimization. We will add an illustration of this to the revised version. **Time Cost**: WoRA introduces three additional trainable parameters compared to LoRA, resulting in a slightly longer training time due to the increased complexity of weight calculation. Although WoRA's training time is marginally higher than that of DoRA and LoRA, the additional trainable parameters enable better weight adjustment, ultimately enhancing performance. As shown in Table 5, a small increase in training time (approximately 30 minutes) significantly improves the model's fine-tuning capability. **Memory Cost**: It is noteworthy that our space complexity is equivalent to DoRA, while WoRA introduces greater freedom in direction and magnitude, thereby facilitating fine-tuning. WoRA adds only two constant learnable scalars per weight update compared to DoRA, resulting in a negligible increase in space complexity.

# 4 EXPERIMENT

## 4.1 Experimental Setup

**Datasets**. Our study involves processing and training on four datasets. For pretraining, we utilize the synthetic dataset, MALS [73], which comprises 1,510,330 image-text pairs, each annotated with relevant attribute labels. For fine-tuning, we employ CUHK-PEDES [37], RSTPReid [86], and ICFG-PEDES. CUHK-PEDES aggregates 40,206 images of 13,003 individuals from 5 existing person reID datasets: CUHK03 [39], Market-1501 [80], SSM [70], VIPER [17], and CUHK01 [38]. Each image is paired with descriptions from two sentences, resulting in a total of 80,412 sentences. Our approach is evaluated on the public text-based person search dataset CUHK-PEDES. RST-PReid contains 20,505 images of 4,101 individuals, compiled from MSMT17 [68]. ICFG-PEDES [13], also derived from MSMT17, consists of 54,522 images of 4,102 individuals.

**Evaluation metrics**. We adopt the mean Average Precision (AP) and Recall@1,5,10 as our primary evaluation metrics. The Recall@K, whose value is 1 if the first matched image has appeared before the K-th image. Recall@K is sensitive to the position of the first matched image and suits the test set with only one true-matched image in the gallery. The average precision (AP) is the area under the PR(Precision-Recall) curve, considering all ground-truth images in the gallery. mAP is calculated and averaged for the average accuracy (AP) of each category.

**Implementation Details**. Our model with the proposed WoRA, undergoes pretraining across 32 epochs on 8 NVIDIA A800 GPUs via Pytorch, adopting a batch size of 150. The optimization strategy employs the AdamW optimizer [46], integrating a weight decay factor of 0.01. Initiation of the learning rate is set at $1e^{-5}$, incorporating a warm-up phase over the initial 2600 steps, which then transitions into a linear decay schedule ranging from $1e^{-4}$ down to $1e^{-5}$. Image preprocessing includes resizing to 384 × 128 dimensions, coupled with augmentation strategies such as random horizontal flipping, RandAugment [11], and random erasing [85]. For the pretraining phase, the image encoder is initialized using Swin-B [45], enhanced by the application of the WoRA method. Both the text encoder and cross encoder commence with configurations derived from the initial and final 6 layers of Bert [12], respectively. Subsequent to the pretraining, the model is fine-tuned on designated downstream datasets over 30 epochs. Initial WoRA settings for the image encoder, rank = 8, $\alpha$ = 8, and $\beta$ = 1, are preserved, with the learning rate commencing at $1e^{-4}$. This phase includes a warm-up period spanning the first three epochs, succeeded by a methodical reduction of the learning rate according to a linear scheduler. In addition to image data augmentation mentioned during pretraining, Easy Data Augmentation (EDA) [67] is employed for text data augmentation, with the batch size as 120. For each text query, its cosine similarity with all images is calculated, selecting the top 128 candidate images. Subsequently, the match probability between the text query and each selected image candidate is computed and ranked [73].

## 4.2 Comparison with Competitive Methods

We deploy the Weighted Low-Rank Adaptation (WoRA) for text-based person search tasks. Performance comparisons are made using Recall@1,5,10 and mean Average Precision (mAP) metrics,

| Method | #Parameter | R@1 | R@5 | R@10 | mAP |
|---|---|---|---|---|---|
| CNN-RNN [53] | - | 8.07 | - | 32.47 | - |
| GNA-RNN [37] | - | 19.05 | - | 53.64 | - |
| PWM-ATH [8] | - | 27.14 | 49.45 | 61.02 | - |
| GLA [7] | - | 43.58 | 66.93 | 76.2 | - |
| Dual Path [83] | - | 44.40 | 66.26 | 75.07 | - |
| CMPM+CMPC [78] | - | 49.37 | - | 79.21 | - |
| MIA [48] | - | 53.10 | 75.00 | 82.90 | - |
| A-GANet [42] | - | 53.14 | 74.03 | 81.95 | - |
| ViTAA [63] | 177M | 55.97 | 75.84 | 83.52 | 51.60 |
| IMG-Net [66] | - | 56.48 | 76.89 | 85.01 | - |
| CMAAM [1] | - | 56.68 | 77.18 | 84.86 | - |
| HGAN [79] | - | 59.00 | 79.49 | 86.62 | - |
| NAFS [16] | 189M | 59.94 | 79.86 | 86.70 | 54.07 |
| DSSL [86] | - | 59.98 | 80.41 | 87.56 | - |
| MGEL [61] | - | 60.27 | 80.01 | 86.74 | - |
| SSAN [13] | - | 61.37 | 80.15 | 86.73 | - |
| NAFS [16] | 189M | 61.50 | 81.19 | 87.51 | - |
| TBPS [18] | 43M | 61.65 | 80.98 | 86.78 | - |
| TIPCB [9] | 185M | 63.63 | 82.82 | 89.01 | - |
| LBUL [65] | - | 64.04 | 82.66 | 87.22 | - |
| CAIBC [64] | - | 64.43 | 82.87 | 88.37 | - |
| AXM-Net [14] | - | 64.44 | 80.52 | 86.77 | 58.73 |
| LGUR [57] | - | 65.25 | 83.12 | 89.00 | - |
| CFine [72] | - | 69.57 | 85.93 | 91.15 | - |
| VGSG [19] | - | 71.38 | 86.75 | 91.86 | 67.91 |
| TBPS-CLIP [5] | 149M | 73.54 | 88.19 | 92.35 | 65.38 |
| IRRA [25] | 194M | 73.38 | 89.93 | 93.71 | 66.13 |
| RaSa [4] | 210M | 76.51 | 90.29 | 94.25 | 69.38 |
| APTM [73] | 214M | 76.53 | 90.04 | 94.15 | 66.91 |
| Baseline* | 214M | 75.42 | 88.86 | 92.77 | 66.61 |
| **Ours** | **127M** | **76.38** | **89.72** | **93.49** | **67.22** |

**Table 1: Performance Comparison on CUHK-PEDES. Here we show the performance of the previous methods on the recall@1,5,10, mAP in % and the parameter number. Baseline*: We re-implement APTM [73].**

alongside a comparison of parameter count (params in Millions, M) and computational efficiency (FLOPs) against the baseline model APTM. Through trainable adjustments to weight parameters, the WoRA method indicated robust performance across both datasets, significantly reducing computational parameters and time. Specifically, compare to APTM, which is trained on 1.51M data, our data filtering algorithm remove 21% of low-quality, noisy data, utilizing 1.19M data for computations. Our implementation of WoRA on the CUHK-PEDES dataset reduce trainable parameters to 127.37M, a 41% decrease from APTM, with FLOPs reduce to 23.21G, a 39% reduction. The overall training duration for pretraining and fine-tuning is cut by 19.82%, with our model achieving slight improvements in recall rates and mAP, as evidenced in Table 1. Moreover, the pretrained model adjusted through WoRA achieves competitive performance on the RSTPReid and ICFG-PEDES dataset, as shown in Table 2 and 3. To show the performance of our model more intuitively, we present three visual retrieval results on CUHK-PEDES

| Method | #Parameter | R@1 | R@5 | R@10 | mAP |
|--------|-----------|------|------|------|------|
| DSSL [86] | - | 32.43 | 55.08 | 63.19 | - |
| LBUL [65] | - | 45.55 | 68.20 | 77.85 | - |
| IVT [58] | - | 46.70 | 70.00 | 78.80 | - |
| CAIBC [64] | - | 47.35 | 69.55 | 79.00 | - |
| CFine [72] | - | 50.55 | 72.50 | 81.60 | - |
| TBPS-CLIP [5] | 149M | 61.95 | 83.55 | 88.75 | 48.26 |
| IRRA [25] | 194M | 60.20 | 81.30 | 88.20 | 47.17 |
| RaSa [4] | 210M | 66.90 | 86.50 | 91.35 | 52.31 |
| APTM [73] | 214M | 67.50 | 85.70 | 91.45 | 52.56 |
| Baseline* | 214M | 66.40 | 85.55 | 91.10 | 52.21 |
| **Ours** | **127M** | **66.85** | **86.45** | **91.10** | **52.49** |

**Table 2: Performance Comparison on RSTPReid. Here we show the performance of the previous methods on the recall@1,5,10 and mAP in % and the parameter number. Baseline*: We re-implement APTM [73].**

| Method | #Parameter | R@1 | R@5 | R@10 | mAP |
|--------|-----------|------|------|------|------|
| Dual Path [83] | - | 38.99 | 59.44 | 68.41 | - |
| CMPM+CMPC [78] | - | 43.51 | 65.44 | 74.26 | - |
| MIA [48] | - | 46.49 | 67.14 | 75.18 | - |
| SCAN [30] | - | 50.05 | 69.65 | 77.21 | - |
| ViTAA [63] | 177M | 50.98 | 68.79 | 75.78 | - |
| SSAN [13] | - | 54.23 | 72.63 | 79.53 | - |
| IVT [58] | - | 56.04 | 73.60 | 80.22 | - |
| LGUR [57] | - | 59.02 | 75.32 | 81.56 | - |
| CFine [72] | - | 60.83 | 76.55 | 82.42 | - |
| TBPS-CLIP [5] | 149M | 65.05 | 80.34 | 85.47 | 39.83 |
| IRRA [25] | 194M | 63.46 | 80.25 | 85.82 | 38.06 |
| RaSa [4] | 210M | 65.28 | 80.04 | 85.12 | 41.29 |
| APTM [73] | 214M | 68.51 | 82.99 | 87.56 | 41.22 |
| Baseline* | 214M | 67.81 | 82.70 | 87.32 | 41.22 |
| **Ours** | **127M** | **68.35** | **83.10** | **87.53** | **42.60** |

**Table 3: Performance Comparison on ICFG-PEDES. Here we show the performance of the previous methods on the recall@1,5,10 and mAP in % and the parameter number. Baseline*: We re-implement APTM [73].**

in Figure 6. Our model adeptly captures fine-grained, word-level details, enabling it to accurately differentiate subtle variations in clothing colors among individuals. Furthermore, it exhibits robust retrieval capabilities, effectively identifying subjects even when parts of their details are obscured. This indicates the strong performance of our model in handling nuanced visual variations and partial occlusions within complex scenes.

### 4.3 Ablation Study and Further Discussion

**The impact of data filtering**. Here, "baseline" refers to the APTM method as implemented in our experimental setup, and "top50" denotes our data filtering approach with a threshold set to top50 for selecting the pre-training dataset MALS. From Table 4, it is evident that training the pre-trained models with our filtered dataset results in improvements of 6.26% in Recall@1, demonstrating the efficacy of our dataset filtering for pre-training. Similarly, we find that filtering on the downstream dataset also facilitates the learning, since the filtering algorithm generally removes the noise. In Table 5,

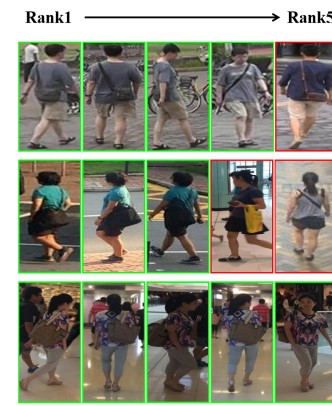

Text Query

The man is facing away and walking in strappy sandals he wears a cross body bag a watch and tan shorts with a gray blue short sleeved shirt he has short dark hair

A woman wearing a light blue shirt a pair of black shorts and a pair of black and brown shoes

She has her long black hair in a pony tail she is also wearing a colorful shirt and light colored pants

**Figure 6: Qualitative person search results using text query of our methods, placing in descending order from left to right based on matching probability. The images in green boxes are the correct matches, and the images in red boxes are the wrong matches. The green texts show that our results successfully match.**

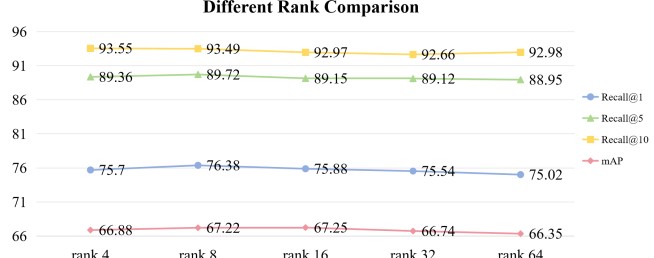

**Figure 7: The impact of different WoRA ranks on performance. We observe that the result is not sensitive to the rank. Generally, rank=8 is the best hyper-parameter in terms of performance on Recall@1,5,10.**

the notation "ft90%" signifies the leverage of our dataset filtering method to refine the finetune dataset CUHK-PEDES, ultimately retaining 90% of its data for training. The model trained after applying "ft90%" exhibits a 0.25% increase in Recall@1, substantiating the effectiveness of dataset filtering on the comprehensive model.

Moreover, we assess the impact of Weighted Low-Rank Adaptation (WoRA) on the complete model. Initially, comparing the impacts of using LoRA [21] and DoRA [43] models of Low-Rank Adaptation on our fully trained model revealed improvements in model speed but not in performance. Subsequently, applying WoRA to models trained on the top50 pre-training selection and the ft90% finetuned dataset, we achieve a Recall@1 of 76.38%, surpassing the baseline by 0.96%, and an mAP of 67.22%, exceeding the baseline by 0.61%, thereby establishing state-of-the-art (SOTA) mAP performance. The computation time for the complete model is reduced from 23h to 18h, marking a 19.82% acceleration. In Figure 7, we compare the effects of different rank values on the performance of WoRA. The experiment shows that the model has the best performance on Recall@1,5,10 when rank is equal to 8. We can observe that model performance may not be as sensitive in different ranks. Performance can be improved basically with the WoRA model. The results we present in this paper are obtained using rank=8, which

| Method | # Data (M) | # Trainable Params (M) | Flops (G) | CUHK-PEDES | | RSTPReid | | ICFG-PEDES | | Time (hours) |
|---|---|---|---|---|---|---|---|---|---|---|
| | | | | R@1 | mAP | R@1 | mAP | R@1 | mAP | |
| *Pretraining Stage* | | | | | | | | | | |
| Baseline* (APTM [73]) | 1.51M | 213.99 M | 38.02 G | 3.99 | 3.62 | 4.40 | 3.95 | 0.77 | 0.59 | 18h |
| Ours (top50) | 1.19M | 213.99 M | 38.00 G | 10.25 | 10.24 | 12.65 | 9.37 | 8.07 | 2.35 | 14h |
| Ours (top50+WoRA) | **1.19M** | **127.37 M** | **23.21 G** | **10.71** | **10.33** | **13.00** | **9.59** | **10.80** | **3.10** | 14h* |
| *Finetune Stage* | | | | | | | | | | |
| Baseline* (APTM [73]) | 0.068M | 213.99 M | 44.93 G | 75.42 | 66.61 | 66.32 | 52.30 | 67.66 | 41.98 | 4.2h |
| Ours (top50) | 0.061M | 213.99 M | 44.93 G | 75.67 | 66.27 | 66.40 | 52.21 | 67.80 | 42.38 | 3.8h |
| Ours (top50+WoRA) | **0.061M** | **127.37 M** | **30.13 G** | **76.38** | **67.22** | **66.85** | **52.49** | **68.35** | **42.60** | 3.8h* |

Table 4: Compared with APTM method at recall@1 and mAP results on CUHK-PEDES, RSTPReid and ICFG-PEDES. Meanwhile, we also compare the data volume, params (M) and Flops (G) of the model. Ablation study about our methods on pretrain. The top50 denotes the results trained by using a pre-trained dataset filtered using a data filtering method. The top50+WoRA is our ultimate two-stage approach, and by adding WoRA to fine-tune the model, we can improve the performance of the pre-trained model while saving 19.82% training time. Time* indicates that the time decrease significantly because the bottleneck of model at other places. Thus, even though we optimize the GPU FPS, it have minimal impact on the overall computation time.

| Method | top50 | ft 90% | R@1 | mAP |
|---|---|---|---|---|
| Baseline* | | | 75.42 | 66.61 |
| Baseline* | ✓ | | 75.83 | 66.70 |
| Baseline* | ✓ | ✓ | 75.67 | 66.27 |
| LoRA | ✓ | | 74.40 | 64.95 |
| LoRA | ✓ | ✓ | 74.29 (-1.38) | 65.59 (-0.68) |
| DoRA | ✓ | | 75.49 | 66.92 |
| DoRA | ✓ | ✓ | 75.73 (+0.06) | 66.75 (+0.48) |
| WoRA (Ours) | ✓ | | 75.67 | 67.09 |
| WoRA (Ours) | ✓ | ✓ | 76.38 (+0.71) | 67.22 (+0.95) |

Table 5: Comparison of our WoRA with baseline, LoRA and DoRA in the different situations. Finally, the experiment shows that our methods achieve the best recall@1 and mAP in %. ft 90% denotes the utilization of our dataset filtering method to refine the finetune dataset CUHK-PEDES, ultimately retaining 90% of its data for training, while top50 is using a data filtering approach with a threshold set to top50 for selecting the pre-training dataset MALS.

has achieved the best performance on Recall@1,5,10. As shown in Table 5, we could observe two points (1) WoRA is better than both LoRA and DoRA, whether the performance of Recall@1 or mAP. (2) Furthermore, if we both leverage the top50 and ft90% filtering strategy, WoRA is also the best performing, which surpasses the baseline by 0.71% Recall@1 and 0.95% mAP under the same conditions. Meanwhile, in order to compare the effectiveness of WoRA. We control all the learning rates unchanged and compare the performance of LoRA, DoRA, and WoRA (ous) in Figure 8. we use the same ranks for LoRA, DoRA, and WoRA for comparison. We can clearly see from the figure that under the condition of a fixed learning rate, the WoRA method proposed by us is consistently better than the LoRA and DoRA model mAP under the condition of the same rank. For example, when the same rank is 64, our method is +0.57% mAP higher than LoRA. +2.38% mAP higher than DoRA.

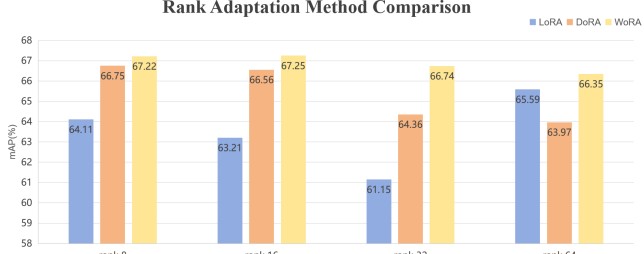

Figure 8: Compare the performance of LoRA, DoRA, and WoRA in terms of different rank. Our WoRA consistently surpasses both LoRA and DoRA on various rank settings.

Therefore, it can be clearly seen that our WoRA method is superior to DoRA and LoRA under the condition of a fixed learning rate.
**The impact of WoRA.** We further investigate the impact of Low-Rank Adaptation on pre-training, where "WoRA" in Table 4 and 5 signifies our Weighted Low-Rank Adaptation approach. Implementing WoRA in the pre-trained models led to a +6.72% boost in Recall@1 and a +6.71% enhancement in mAP, concurrently reducing the time by 22.22%, thereby validating the efficiency of our dataset filtering and WoRA in model pre-training.

## 5 CONCLUSION

In this work, we introduce a new Filtering-WoRA paradigm, which contains a filtering algorithm to identify this crucial data subset and WoRA layers (Weighted Low-Rank Adaptation) for light fine-tuning. Filtering strategy for image-text pairs within language-based person search datasets, designed to isolate a core set from large-scale, noise-containing datasets of generated image-text pairs. WoRA (Weighted Low-Rank Adaptation) learning strategy to efficiently update the portion of model parameters. Extensive experiments indicate that our approach is 19.82% faster than existing language-based person search methods while maintaining comparable accuracy with state-of-the-art (SOTA) language-based person search models. On three public benchmarks, CUHK-PEDES, RSTPReeid, and ICFG-PEDES, our method achieves competitive recall rates and mean Average Precision (mAP).

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
