# OpenReview forum: "From Data Deluge to Data Curation: A Filtering-WoRA Paradigm for Efficient Text-based Person Search"
_ACM.org/TheWebConf/2025/Conference — WWW 2025 Poster_

### Official Review · Reviewer_z3rp · 2024-11-24

**Novelty:** 6
**Technical Quality:** 6

**Review:**

### Summary:
This paper defines the problem that the quality of training data is closely tied to the performance of person search models, which often have strict latency requirements. Therefore, how to identify a subset of training data rich in useful information from a large dataset? How to significantly reduce training time and model complexity? These are the key issues to address. To solve the first problem, the authors propose a novel two-stage data filtration method. To address the second problem, they introduce WORA, a new fine-tuning method. Comprehensive and extensive experiments demonstrate the effectiveness of the proposed approach, making this appear to be a high-quality paper.
### Pros:
1. The writing of the paper is very clear. For example, Figure 2 clearly illustrates that the method consists of Data Filter and WoRA, Figure 4 provides an intuitive demonstration of the effectiveness of Data Filter, and Figure 5 offers a clear mathematical explanation of WoRA.
2. The problem is rigorously defined in the paper, with a well-structured description that makes the challenges very clear.  The authors also provide quantitative evidence to validate the existence of the problem, effectively demonstrating its significance through experimental results.
3. The paper has considerable practical value, particularly as it considers key concerns such as retrieval latency and the potential future scenarios in person search, making it relevant to both academic research and real-world applications.
4. The experimental results are impressive, as the model maintains high performance with fewer parameters compared to the baseline.
### Cons:
1. The paper does not provide a direct comparison between the proposed Data Filtering method and other existing data selection techniques. Including such comparisons would provide a clearer understanding of the strengths and limitations of the proposed approach. Additionally, an exploration of how different data selection strategies impact the final performance could strengthen the overall argument for the effectiveness of the authors' method.

**Questions:**

1. It would be helpful to better illustrate the filtering pattern of the proposed method by including an experiment on the impact of the filtering ratio on data quality. Do the authors have any experimental results regarding this aspect?
2. Is the Data Filtering method time-consuming? As I understand, it is part of your pre-training preparation, and its duration would also impact the overall training time.

**Reviewer Confidence:**

3: The reviewer is confident but not certain that the evaluation is correct

**Scope:**

4: The work is relevant to the Web and to the track, and is of broad interest to the community

---

### Official Review · Reviewer_51Sw · 2024-11-29

**Novelty:** 4
**Technical Quality:** 4

**Review:**

The paper proposes a novel Filtering-WoRA paradigm for efficient text-based person search. It introduces a two-stage method: a filtering algorithm to curate high-quality, relevant subsets from large-scale datasets and the WoRA learning strategy for lightweight fine-tuning. By leveraging cross-modality relevance and low-rank weight adaptations, the method reduces computational overhead and training time while achieving competitive performance.

### Strengths
1. The Filtering-WoRA paradigm combines effective data curation with a lightweight adaptation strategy, addressing both data quality and computational efficiency in training large models.
2. Extensive experiments show that the method reduces training time by 19.82% while maintaining or improving retrieval performance, highlighting its practical value for resource-constrained applications.
3. The proposed methods are applicable to multiple datasets and tasks, showcasing their robustness and transferability across different scenarios.

### Weaknesses
1. WoRA, as a Parameter-Efficient Fine-Tuning method, is primarily tested in the specific domain of text-based person search. While it demonstrates promising results in this context, the paper does not provide sufficient validation of the method across a broader range of tasks or domains。
2. The experiments in the paper focus on relatively smaller base models, but WoRA's effectiveness as a PEFT method is not validated on models with significantly larger parameter sizes. It is unclear how WoRA performs when applied to large-scale models, such as those with billions of parameters, which are commonly used in state-of-the-art NLP and CV tasks.

**Questions:**

1. Could you elaborate on how the thresholds for filtering (e.g., top 50, top 90%) were determined, and whether these settings are dataset-specific?
2. Have you tested the scalability of the Filtering-WoRA paradigm with larger models, and if so, what are the outcomes and limitations?
3. How does your filtering algorithm compare quantitatively and qualitatively to existing dataset curation techniques, such as active learning or coreset selection methods?

**Reviewer Confidence:**

2: The reviewer is willing to defend the evaluation, but it is likely that the reviewer did not understand parts of the paper

**Scope:**

3: The work is somewhat relevant to the Web and to the track, and is of narrow interest to a sub-community

---

### Official Review · Reviewer_fQWU · 2024-12-02

**Novelty:** 5
**Technical Quality:** 6

**Review:**

The authors propose the Filtering-WoRA paradigm to address the efficiency of text-based person search, especially during the training phase. The approach includes a filtering algorithm (to retain only a core subset of high-quality training data) and the adaptation method WoRA for lightweight fine-tuning. By introducing alpha and beta parameters for direction and amplitude, the authors aim to enhance flexibility and adaptability while reducing computational costs. The work builds on state-of-the-art methods and seeks to outperform existing LoRA and DoRA algorithms.

Positive point is to improve methods for reducing noise in training data and minimizing the overall volume of training data through meaningful filtering. The approach ensures that the model trains on the most relevant data without requiring manual fine-tuning as the dataset expands. It is credible that alpha and beta parameters provide the model with greater flexibility and improve adaptability to datasets similar to those used in training. These points convincingly contribute to achieving state-of-the-art results while reducing training time (and the experiments look good).

However, I have some concerns about potential overfitting to the filtered subset, which might exclude rare but important features. This limitation could result in a loss of critical diversity, particularly in edge cases, which were not thoroughly analyzed in the paper (nor in the SOTA methods it builds upon, making this less crucial in comparison). Also, though the approach seems to be effective, the method is rather simple introduction of parameters to DoRA which can not be considered to be especially novel

Overall, the paper effectively improves SOTA in terms of efficiency and effectiveness, as demonstrated in extensive experiments. Further exploration of edge cases  could enhance its robustness and applicability, but the work as presented is a solid contribution to text-based person search.

**Questions:**

Could the beta parameter simply be replaced by beta = (1-alpha), potentially simplifying the model without loss of functionality?
Could you present also losses the filtering method is introducing (e.g. on some edge cases)

**Reviewer Confidence:**

2: The reviewer is willing to defend the evaluation, but it is likely that the reviewer did not understand parts of the paper

**Scope:**

4: The work is relevant to the Web and to the track, and is of broad interest to the community

---

### Official Review · Reviewer_Rut4 · 2024-12-03

**Novelty:** 3
**Technical Quality:** 3

**Review:**

The paper proposes a method to improve model effectiveness and training efficiency in cross-modal person search applications. The authors suggest methods to filter out low-quality data points and train on a core dataset to enhance model training effectiveness. They also propose updating partial parameters to improve training efficiency.

**Pros:**
The experimental section includes comprehensive baselines and multiple ablation experiments, which provide insights into the performance of the proposed method.

**Cons:**
1. The novelty of this paper is unclear. The filtering algorithm appears relatively simple, and the partial parameter updating mechanism offers only a slight improvement over existing methods.
2. It is unclear why the authors limit the application to the person search scenario. What unique qualities of person search are absent in other text-image search scenarios?
3. The experimental results do not show significant improvement over the baselines.

**Questions:**

1. Filtering down to a core dataset and updating partial parameters seem like two separate improvements that can be applied to many machine learning solutions. The overall improvement proposed in this work feels incremental and may not be significant enough to warrant a full paper.
2. The authors could consider evaluating the quality of the core dataset generated by their algorithm.
3. The first paragraph of Section 3.3 seems more appropriate for the related work section.
4. Please include significance tests for Tables 1 through 5.
5. Table 4 reports a 4-hour training improvement in the pre-training stage and a 0.4-hour improvement in the fine-tuning stage. These gains do not seem meaningful compared to the total time required for each stage.
6. The R@1 and MAP improvements in Tables 4 and 5 are minimal, further suggesting that the contributions of this work are incremental.

**Reviewer Confidence:**

4: The reviewer is certain that the evaluation is correct and very familiar with the relevant literature

**Scope:**

3: The work is somewhat relevant to the Web and to the track, and is of narrow interest to a sub-community